# *Drosophila melanogaster* Systemic Infection Model to Study Altered Virulence during Polymicrobial Infection by *Aeromonas*

**DOI:** 10.3390/pathogens12030405

**Published:** 2023-03-02

**Authors:** Alexandre Robert, Emilie Talagrand-Reboul, Maria-Jose Figueras, Raymond Ruimy, Laurent Boyer, Brigitte Lamy

**Affiliations:** 1Laboratoire C3M, Inserm U1065, Université Côte d’Azur, 06200 Nice, France; 2Service de Médecine Intensive et Réanimation, Pole Anesthésie/Réanimation/Bloc Opératoire, Centre Hospitalier de Cannes, 06400 Cannes, France; 3Laboratoire de Bactériologie, Pole Biologie, CHRU de Strasbourg, UR7290, Université de Strasbourg, 67000 Strasbourg, France; 4Unitat de Microbiologia, Departament de Ciènces Médiques Bàsiques, Facultat de Medicina i Ciències de la Salut, IISPV, Universitat Rovira i Virgili, 43201 Reus, Spain; 5Laboratoire de Bactériologie, Pole Biologie/Pathologie, CHU de Nice, Université de Nice Côte d’Azur, 06200 Nice, France; 6MRC Centre for Molecular Bacteriology and Infection, Imperial College of London, London SW7 2AZ, UK

**Keywords:** polymicrobial infection, *Drosophila melanogaster*, infection model, opportunistic pathogen, *Aeromonas*, microbial interference

## Abstract

Background: Polymicrobial infections are complex infections associated with worse outcomes compared to monomicrobial infections. We need simple, fast, and cost-effective animal models to assess their still poorly known pathogenesis. Methods: We developed a *Drosophila melanogaster* polymicrobial infection model for opportunistic pathogens and assessed its capacity to discriminate the effects of bacterial mixtures taken from cases of human polymicrobial infections by *Aeromonas* strains. A systemic infection was obtained by needle pricking the dorsal thorax of the flies, and the fly survival was monitored over time. Different lineages of the flies were infected by a single strain or paired strains (strain ratio 1:1). Results: Individual strains killed more than 80% of the flies in 20 h. The course of infection could be altered with a microbial mix. The model could distinguish between the diverse effects (synergistic, antagonistic, and no difference) that resulted in a milder, more severe, or similar infection, depending on the paired strain considered. We then investigated the determinants of the effects. The effects were maintained in deficient fly lineages for the main signaling pathways (Toll deficient and IMD deficient), which suggests an active microbe/microbe/host interaction. Conclusion: These results indicate that the *D. melanogaster* systemic infection model is consistent with the study of polymicrobial infection.

## 1. Introduction

A polymicrobial infection is a complex situation that involves two or more microorganisms and a host. This system is influenced by microbial population dynamics; interspecies microbial interactions that result in competition, cooperation, or neutrality; and interactions with the host immune system, including immune modulation or manipulation [1,2,3]. These multifaceted interactions impact the infection outcome, which arises from a more severe infection, in relation to a synergistic effect on the infection outcome, to a less severe infection or even infection avoidance, in relation to an antagonistic effect [1,2,3,4]. Polymicrobial infections are related to an increasing burden [4]. They are encountered during chronic infection, e.g., diabetic foot infection or chronic infection in the cystic fibrosis lung, but also during acute infections, e.g., peritonitis, gastroenteritis, ventilator-associated pneumonia, deep-seated abscesses, and bacteremia [4,5,6]. Importantly, they are frequently associated with generalist opportunistic pathogens for which the infectiveness cannot be predicted from the genome content of virulence-associated genes and for which the environment and the polymicrobial condition modulate the infectiveness [7,8,9,10]. The outcomes are often dependent on the strain properties [11,12].

Understanding the complex interactions that occur during a polymicrobial infection is critical to advance the knowledge of infection pathophysiology. This knowledge can help with the development of novel strategies for remedial treatments, adjunct treatments, and prophylactic treatments or pathogenicity control, which are important given the lapsing effectiveness of conventional antimicrobial therapies [1,13]. Polymicrobial infection modeling gains real importance within this context. Many studies have focused on microbe–microbe relationships in in vitro studies, which is a truncated approach to capture the complexity of infectious processes. These studies ignore the host contributions, damage–response frameworks, and/or microbes that can interact differently inside hosts [14,15]. Instead, an in vivo infection comprises growth dynamics thwarted by the host innate immunity, possible changes in the host danger sensing, and innate immunity manipulation [1,9,16,17].

An important step is to construct easy-to-perform models that efficiently discriminate between polymicrobial-based effects on the infection outcome, i.e., antagonistic effects, synergistic effects, and neutral effects, compared with monomicrobial infection. Several models exist, but many have a long time to results and are costly or troublesome regarding animal welfare. Recently, a *Drosophila melanogaster* model generated by needle pricking the dorsal thorax of flies was used to study polymicrobial infection [18]. It is powerful because many virulence mechanisms are active across a wide spectrum of hosts, innate immune response pathways are evolutionarily conserved, and many opportunistic human pathogens have been successfully studied using this system [19,20]. The model is fast and allows the infection of a high number of animals, the outputs are easy to achieve (e.g., animal survival, bacterial load, and gene expression, for example), and various tools such as mutant and transgenic animals or microscopic platforms allow the study of the components of the infection (e.g., bacterial aggregation or dissemination, microbe dynamics, and antimicrobial effectors or cell immunity). Such in vivo simplified platforms are promising to uncover some microbe–microbe and host–microbe interactions and may contribute to develop new therapeutic concepts to reduce bacterial infectiveness as long-term goals. Notably, the *D. melanogaster* systemic infection model was exemplified with a case of synergistic infection by a bacterial mix. However, no evidence was shown that the model was able to discriminate between the paired strains that cause a more severe infection and those that cause a less severe infection or a similar infection [18].

In this work, we used a *D. melanogaster* infection model to assess whether it can accurately discriminate between the synergistic and antagonistic effects during a polymicrobial infection, as a first milestone to further investigate polymicrobial infection. Flies were infected with aeromonads. These bacteria are a good model of ubiquitous opportunistic pathogens [21,22] with a wide range of hosts, including insects [23], and whose pathogenicity is influenced by polymicrobial conditions [9,12]. Their pathogeny, especially in humans, is poorly understood, and the content of the virulence-associated genes can neither predict the virulence of strains nor highly severe infection [24]. The companion strain is thought to modulate the aeromonad virulence [9,12,25], supported by a frequent polymicrobial culture in a clinical situation, including with another *Aeromonas* strain [24,25].

## 2. Materials and Methods

### 2.1. Strain Collection and Culture Conditions

Three different paired bacterial strains (six strains) were recovered from clinical specimens (Table 1). The strain (28b) was recovered as a single isolate. All aeromonad strains were identified at the species level by whole genome sequencing, as previously described [26]. The genome content of virulence-associated genes was explored in a previous work (European Nucleotide Archive database accession numbers: PRJEB8966, PRJEB9012, PRJEB9013, PRJEB9014, PRJEB9016) [12]. Bacteria were kept frozen at −80 °C in brain heart infusion broth with 20% glycerol until analysis. They were grown on Lysogenic Broth (LB) agar (MP Biomedicals) overnight at 37 °C, and single colonies were then cultured in LB broth overnight in a shaking incubator (180 RPM) at 37 °C. A subsequent subculture was performed by inoculating LB broth with 1/1000 diluted overnight LB culture and incubated for 12–16 h at 37 °C. Growth curves were performed using a microplate reader (TECAN infinite 200) over 12 h, and all paired strains showed similar growth rates, either grown individually or in pairs (Appendix A). Strains were studied alone or in combination with their natural co-isolates (Table 1).

### 2.2. Drosophila melanogaster Infection Model

Briefly, cultured broths were adjusted to an optical density (600 nm) of 1.0 (approx. 10^8^ CFU/mL) using the media the strain was grown in. This standardized suspension was used to prepare inoculums for infection, as follows. For the single-strain inoculums, 1 mL of the suspension was collected and centrifuged at 17,500× *g* for 2 min 30 s. For paired-strain inoculum, an equal volume of the standardized suspension of each strain was collected and mixed, using a ratio of 1:1 according to previous work [12]. The suspension was centrifuged, the supernatant was discarded, and the pellet was resuspended in 70 μL of PBS before fly pricking.

Preliminary experiments were led using two inoculums at a ratio of 1:1, prepared from the standardized suspensions: 0.5 mL + 0.5 mL (hereafter mix A) and 1 mL + 1 mL (hereafter mix B) (Figure 1A). Mix A was designed to compare the outcomes of monomicrobial and polymicrobial infections with same total amount of bacteria (all cells cumulated) compared to single-strain suspensions. Mix B was designed to compare the same amount of each individual microorganism compared to single-strain suspensions, resulting in a 2-fold total number of cells (Figure 1A). Mix B inoculum was used for further experiments for infections with a pair that comprised a nonpathogenic bacterium (e.g., strain *E. coli* [27] for the pair 186/187). It was also used in case of an antagonistic effect, to avoid artefactual effect that would be related to different inoculum of either bacterium in the mix. A mixed inoculum that comprised one dead bacterium was prepared as described above after one of the standardized inoculums was heated at 70 °C for 30 min. These were used as control infections. A mixed inoculum that comprised all dead bacteria after they were grown together was prepared similarly.

A total of 30 adult (3–5 days old) male flies grown on cornmeal medium under standardized conditions. Flies were infected by pricking the thorax of a CO_2_ anesthetized fly with a 0.4 mm tungsten needle dipped into the bacterial suspension, which resulted in an approx. inoculum of 500 CFU in the fly body. At least three independent experiments were carried out. Assays were performed using three lineages of *D. melanogaster*: (1) wild-type lineage White (WT, w1118); (2) IMD-pathway deficient flies with the inactivated transcription factor Relish gene (DmRel w1118; RelE20) provided by Bloomington *D. melanogaster* Stock Center; and (3) Toll pathway-deficient flies with the inactivated adaptor Myd88 gene (DmMyd88, c03881), a kind gift from JL Imler (Institut de Biologie Moléculaire et Cellulaire, Strasbourg, France). The control group corresponded to flies pricked with *Escherichia coli* strain 187, a strain that belongs to a nonvirulent species in a *D. melanogaster* wild-type infection model [19]. *Aeromonas dhakensis* strain 28b, known to be highly virulent, was used as a positive control [12]. After inoculation, flies were housed at 25 °C, and survival was assessed every 2 h for 24 h. The effect of the polymicrobial condition was considered to be antagonistic when fly killing was slower with the mix than with the most virulent pair strain tested individually, using inoculum mix B, i.e., slower killing despite the mixed inoculum containing the same amount of the most virulent strain. The effect of the polymicrobial condition in the wild-type fly was considered to be synergistic when fly killing was faster with the mix than with the most virulent strain pair tested individually, using inoculum mix A, i.e., faster killing despite the mixed inoculum containing the same total amount of bacteria and half the amount of the most virulent strain.

### 2.3. Quantitative Microbiology

At T0 post infection, three flies were homogenized in Phosphate buffer saline (PBS) and serially diluted in LB. Dilutions were cultured on LB agar plates overnight at 37 °C under aerobic conditions. Colony-forming units (CFU) were counted to enumerate the bacterial load, CFU/fly, and the inoculum ratio was determined for infection conditions with mixed bacteria. Gut commensal bacteria (mostly anaerobic) were excluded upon morphological aspect (e.g., colony size < 0.3 mm) due to their limited growth compared to *E. coli* and aeromonads under these culture conditions [28,29]. In the case of the bacterial mix, direct colony counts of each bacterium were performed based on the physical/visual differences in the colonies at day 2 post culture thanks to clear differences (Appendix A). Similar counts and inoculum ratio determination were performed at T0 post infection on suspensions used for fly infection. Total bacterial burden in the fly was evaluated just before flies started dying (H6). To do this, eight flies were individually homogenized, serially diluted in PBS, and cultured as described above.

**Table 1 pathogens-12-00405-t001:** Strain characteristics used in this study and context of strain isolation.

Strains Characteristics					Context of Strain Isolation				
Species	Strain	Region, Country, Year of Isolation	Pair	Origin	Patient No.	Type of Infection	Age/Sex	Comorbidities and Predisposing Factors	Comments
*A. rivipollensis* *	76c	Barcelona, Spain, 1992	1	Stool	1	Diarrhea with fever and unremarkable presentation	5/M	Hydrocephalus and ventricular-peritoneal shunt, catheter-related coinfection	Aeromonads considered by the clinician as responsible for the diarrhea after no other pathogen was isolated during the stool culture bacterial screening
*A. veronii*	77c	Barcelona, Spain, 1992	1	Stool					
*A. hydrophyla*	25a	Saint-Brieuc, France, 2006	2	Respiratory tract	2	No infection (colonization)	62/M	Bronchiectasis Corticosteroid treatment [30]	Clinician decision not to treat [30]
*A. veronii*	25b	Saint-Brieuc, France, 2006	2	Respiratory tract					
*A. veronii*	186	Montpellier, France, 2015	3	Blood	3	Bloodstream infection with septic shock	72/M	Cardiopathy Chronic kidney failure History of cholecystectomy following a large cholelithiasis with a resulting blind loop of the bowel	Clinician decision to administer aeromonad-targeted antimicrobial treatment
*E. coli*	187	Montpellier, France, 2015	3	Blood					

* This strain used in previous study [12] under the name *Aeromonas media* was later recognized by [31] to belong to the newly defined species *A. rivipollensis* [32].

### 2.4. Statistical Analysis

Data analyses were performed using GraphPad Prism version 5.00 for Windows (GraphPad Software, San Diego, CA, USA). The Kaplan–Meier method was used to plot survival curves. Comparisons of survival curves obtained within each experiment were performed using log-rank tests to determine whether the survival observed with the paired strains differed from those of single strains. Comparisons were performed between the curve of the mix and that of the individual strain that exhibited the fastest fly killing. A *p*-value ≤ 0.05 was considered to reflect significance. Curve comparisons were independently performed for every assay. Two conditions were required to establish antagonism or synergy: (i) for a given assay, survival curve from the mixed infection was significantly slower or faster (hereafter effect) compared to that of the monomicrobial infection, respectively, and (ii) this effect was reproducible, with a significant difference found from at least 3 independent assays. Antagonism definition also required that assays include the same amount of each individual microorganism compared to single-strain suspensions.

## 3. Results

### 3.1. Characteristics of the Infections

The paired strains were recovered from three different types of clinical situations. (Table 1). One was a systemic infection (gut-associated bloodstream infection) that involved the *E. coli* strain 187 and *Aeromonas veronii* strain 186 in a patient who had several comorbidities; one was gastroenteritis caused by two distinct aeromonads (strains 76c and 77c) in a young patient with several underlying diseases (Table 1); and one was an untreated colonization of the respiratory tract by two aeromonads (strains 25a and 25b) in a patient who suffered from a pulmonary underlying disease.

### 3.2. Characteristics of Individual Strain Virulence

Wild-type (WT, w1118) *D. melanogaster* were infected with an average inoculum of 530 ± 100 CFU per fly. Single-strain assays in the wild-type (WT, w1118) *D. melanogaster* systemic infection model showed diverse virulence profiles (Appendix A). Individual strains 186, 77c, 25a, and 25b killed >80% of the flies in 10 to 20 h, while strain *Aeromonas* 76c was less virulent (≤50% of the flies were killed at H 20 p.i.). The *E. coli* strain 187 was avirulent to wild-type (WT, w1118) *D. melanogaster* (Lemaitre and Hoffmann, 2007) [27].

### 3.3. Assessing the Effect of Polymicrobial Situation on the Infection Outcome

We then investigated the effect of a polymicrobial situation by pairing the strains recovered from a clinical infection, as detailed in Table 1. The flies were infected with an average total inoculum of 430 ± 90 CFU/fly for mix A and 1190 ± 200 CFU/fly for mix B. The variability in the ratio inoculum was between 1:1 and 1:2 for 10 out of the 12 experiments, and none exceeded 1:3. The type of effect was reproducible with this range of conditions.

Three types of effects were observed. First, *A. hydrophila* 25a and *A. veronii* 25b exhibited similar virulence phenotypes when tested in pairs or individually (*p*-values between 0.4 and 0.5, Figure 1B,C). Second, the addition of the avirulent *E. coli* strain 187 to *A. veronii* strain 186 resulted in a slight but significant synergistic effect (*p*-values between 0.046 and 2 × 10^−4^, Figure 1C). Finally, we found an antagonistic effect on the infection outcome with the mixture of strains 76c and 77c because the virulence of the pair was significantly lower than that of the most virulent strain of the pair tested individually (strain 77c) (*p*-values between 0.05 and <10^−4^, Figure 1C).

With a neutral effect (pair 25a/25b), there was a 10-fold difference between the counts of the two single-strain conditions, while the count of the mix was similar to that of 25b (Figure 1D). With the synergistic effect, the median bacterial burden in the fly at the time point six hours post infection (hereafter H6) were similar to the individual strains (NS). The mix and strain 186 exhibited a similar median bacterial burden at H6 while the count of the nonpathogenic strain 187 was 10-fold higher (Figure 1D). With the antagonistic effect, a slight but insignificant (*p*-value 0.11) reduced bacterial burden was observed with the mix at H6 post infection compared with that of strain 77c (Figure 1D) while the individual strain 76c was cleared and strain 77c grew in the fly. These results suggest that a polymicrobial infection is associated with complex mechanisms.

### 3.4. Screening Bacterial Relationships during Polymicrobial Infection in the Host

We further explored pairs that showed a difference in the infection outcome relative to the individual strain infection (pairs 76c/77c and 186/187). To investigate the role of the companion strain on the infection outcome, we tested the effect of a mix that comprised one living strain and one dead strain on wild-type (WT, w1118) *D. melanogaster.* The resulting survival curve was similar to that of the individual living bacterium with 76c, 77c, 186, or 187 (Figure 2). In addition, we infected flies with a mix that comprised the two bacteria grown together and killed before infection, which resulted in 100% fly survival (Figure 2).

### 3.5. Exploring the Role of Innate Immunity during Polymicrobial Infection

To explore the role of the two principal signaling pathways of the immune system of the fly, infection experiments were performed in parallel on mutant lineages of *D. melanogaster* using Toll-deficient flies and IMD-deficient flies. The IMD pathway is a signaling cascade activated in response to the signals produced by Gram-negative bacteria through peptidoglycan recognition proteins (PGRPs) and results in the translocation of the NFκB-like transcription factor Relish, which leads to antimicrobial peptide production. Toll receptors are conserved components of the immune system, and MyD88 is an adaptor protein in the hToll/IL-1 receptor family signaling cascade that also uses the NFκB pathway. The effect of the polymicrobial condition on the infection was maintained in both deficient fly lineages with the pair 76c/77c and the pair 186/187 (Figure 3).

## 4. Discussion

Polymicrobial conditions are increasingly recognized as influencing infection outcomes in relation to the amazing versatility of microbes [6,25,32]. Despite some advances, little is known about the complex mechanisms involved and calls for efficient models to better decipher these infections. Model organisms have emerged as relevant in vivo models to study infection pathophysiology of polymicrobial infections [11,12,33,34], of which the fruit fly model is one. Our results add to the existing evidence the fly model is a promising model to study polymicrobial infection [8,11,35].

### 4.1. Drosophila Melanogaster, a Model Organism to Study Polymicrobial Infection

*D. melanogaster* models for polymicrobial infection were first developed to assess the ability of streptococci from cystic fibrosis airways to influence the course of infection when combined with *Pseudomonas aeruginosa* or to study the interaction between *P. aeruginosa* and *Staphylococcus aureus* [8,11]. Both studies used orally infected flies, a model limited by the challenge to control the amount of each microorganism ingested by the fly. Instead, the *D. melanogaster* systemic infection model enables a better standardized inoculum and was recently applied to polymicrobial infection [18], albeit its ability to distinguish between various types of effects was not documented. Here, we show that it can reproducibly discriminate between synergistic, antagonistic, or neutral effects related to a polymicrobial situation. These observations may be consistent with the severity observed in humans (e.g., systemic vs. non-systemic infection), even though they may also be coincidental. Our results suggest some complex mechanisms behind these observations: first, these effects cannot be related to a particular virulence profile in aeromonads [12]; the effects require living bacteria which suggests an active microbe/microbe or microbe/host interaction; and the effects are maintained when the principal signaling pathways to activate the host innate immunity, TOLL and IMD, are inactivated. While this mutant data could be reinforced with assays using the RNAi lines of these flies, they suggest that either none of the TOLL and IMD signaling pathways is involved in the modulation of infection or, if one is, it involves an effector that is blocked by the inactivation of neither the adaptor Myd88 nor the transcription factor Relish. Instead, it may involve some additional factor of fly immunity that requires further investigation.

### 4.2. Characteristics, Possibilities, Limitations, and Weaknesses of the Fly Infection Model

The model was set up using a high inoculum and a ratio of 1:1 on the basis of previous works [9,12,36], although it can be adapted to the bacterial mix studied. The flies offer many possibilities for further investigations. The assets include mutant and transgenic animals, a quantitative (host and pathogen) gene expression analysis in the fly, labeling-based microscopy to study bacterial localization and population dynamics, assays to assess humoral and cellular responses, histopathology, or even fly behavior. The model allows the study of the components of virulence (e.g., population dynamics, bacterial fitness in vivo, and the type of host defense involved) that are important during the acute invasion phase. Previously, the fly model was used to assess the virulence of a mix recovered in post-surgery flap infection-associated sepsis following leech therapy, a situation that causes rapid flap loss. While it is poorly understood why the flap loss occurs so rapidly, the synergy that was observed between the two aeromonad isolates may explain at least in part this rapid loss [25]. In this study, the model fitted well to the conditions of infection: flap immunity relies on innate immunity following locally immunocompromised tissues as a result of impaired vascularization, and the leech creates a wound with its jaws and regurgitates aeromonads into the flap, similar to an injection.

The potential limitations of the fly systemic infection model include a low throughput, an animal size that limits some experiments, or relative variability in the inoculums that is less important than that in the ingestion model. In this study, the total and relative inoculum exhibited some variability. It originates from the complexity of the model (one host, two bacteria), high inoculums, and the use of the pricking method. Indeed, the variability in the administered inoculum is a known limitation of the pricking method. Using a fly nanoinjector to cause infection instead of a needle significantly reduces the inoculum variability between animals, but it is at the expense of heavier equipment [16,37]. Despite a more variable inoculum with the pricking method, it is acknowledged that the pricking method provides consistent results [18,38,39]. In this study, we found reproducible effects (neutral, antagonistic, and synergistic). Another limitation is that the injured fly model is, to date, poorly adapted to study complex microbial shift diseases (e.g., vaginosis) [2] or the effects of microbial biofilms inside the host (e.g., cystic fibrosis lung).

### 4.3. The Polymicrobial Condition Modulates the Pathogenicity of Opportunistic Bacteria

We show that the course of infection caused by aeromonads is affected by a polymicrobial condition, as a result of interactions with another microbe or host that alter its pathogenicity. Fast bacterial adaptation is required to achieve this switch, which is consistent with the high and rapid adaptation capabilities of aeromonads to their altered surroundings [7,40]. A polymicrobial situation is frequent and may contribute to the pathogenicity of opportunistic pathogens through microbial interactions, as reported elsewhere [8,9,10,12,21,30,41,42,43].

We observed a mild synergy when strain 186 was mixed with its companion strain *E. coli* 187, a strain that is individually avirulent to the fruit fly [27]. Although mining the mechanisms related to this effect was out of the scope of this preliminary study, we found that a mix does not necessarily need to include pathogens to be harmful, consistent with Sibley et al.’s data [11]. In Sibley et al.’s study, Canton S flies were orally infected with oropharyngeal isolates from cystic fibrosis (CF) sputum, either individually or with *Pseudomonas aeruginosa*. Many organisms (e.g., streptococci, staphylococci, and actinomyces) were avirulent to the flies when ingested alone but reduced fly survival when they were ingested with *Pseudomonas aeruginosa*. This suggests that many organisms have the ability to influence the course of an infection when combined with a pathogen [11]. We show that the two bacteria need to be alive to produce that synergy, and the data suggest either an interference competition between strains or a microbial interaction that triggers a less effective immune response that is regulated by neither Myd88 nor Relish.

Interestingly, the 76c/77c pair had an antagonistic effect that is associated with a less severe infection, an effect that is seldom reported. While ecological and evolutionary theory generally predicts microbes cause more damage in multispecies infections than in single-species infections [10], it also predicts that polymicrobial situations cause less damage for several well-defined conditions. Among these situations, a high level of competition over shared virulence factors between virulence factor producers and nonproducers (cheaters) is predicted to attenuate host damage because the overall amount of virulence factors decreases [44,45]. Another well-defined condition for which lowered virulence is predicted to occur is when specific toxins aimed at killing competitors are produced by one of the microbes or indirectly once host immunity has been activated by one of the bacteria [10]. While most studies focus on colonization resistance in undamaged barrier environments, there are, to our knowledge, very few reports of mitigated pathogenicity during a polymicrobial infection in progress. One study concerns a complex interplay between the host and aeromonads that involve secretion systems [9]; another concerns a diabetic foot ulcer colonized by *Helcococcus kunzii* that decreased the virulence of *S. aureus* during infection without directly modifying the host defense response [32]; and another concerns a *Morganella morganii* strain that secretes products that decrease the *Proteus mirabilis* urease activity, a factor associated with a more severe catheter-related urinary tract infection [6].

## 5. Conclusions

This study confirms the ability of the *D. melanogaster* systemic infection model to study nuanced infection modulation during a polymicrobial infection and to discriminate between the synergistic and antagonistic effects. The suitability and discriminatory power now allow the use of the model for large-scale prescreenings to study strain pairings and perform a mechanistic investigation thanks to the numerous tools available in the *D. melanogaster* model. The well-researched fly immune response and its similarity to the mammalian innate immune system make the fruit fly an attractive and good stepping-stone model to monitor the degree of pathogenicity that occurs during a polymicrobial infection in vivo.

## Figures and Tables

**Figure 1 pathogens-12-00405-f001:**
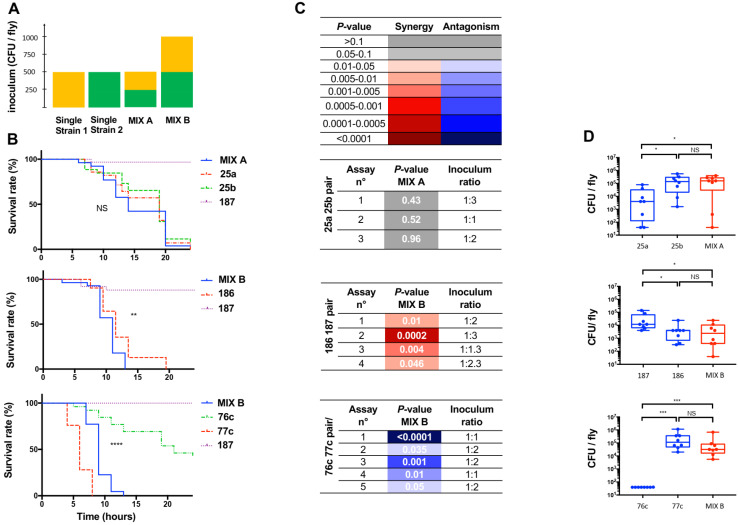
Characteristics of polymicrobial and monomicrobial *Drosophila melanogaster* wild-type (WT, w1118) infection by pricking with three clinical pairs. The survival curve of coinfection was compared to the survival curve of infection with the individual strain that exhibited the fastest fly killing. *n* = 30 flies per condition, target ratio of 1:1, *E. coli* strain 187 used as a negative control, average inoculum of either 500 or 1000 CFU/fly. Experiments were carried out with a mix A inoculum to ensure comparable inoculum between monomicrobial and polymicrobial conditions, ± a mix B inoculum in case of an antagonistic effect on the infection outcome or when one of the bacteria was avirulent (*E. coli*) to ensure comparable inoculum of the strain that exhibited the fastest fly killing between monomicrobial and polymicrobial conditions. (**A**) Characteristics of mix A and mix B, as detailed in the method section. (**B**) Survival curves of flies infected with three clinical pairs, strains 25a ± 25b (top), strains 186 ± 187 (middle), and strains 76c ± 77c (bottom). Survival curves refer to results of one replicate and are representative of at least three independent experiments of the strains tested individually (dotted lines) and in pairs (solid lines), **** *p*-value < 0.0001, *** *p*-value between 0.001 and 0.0001 *, ** *p*-value between 0.01 and 0.001, * *p*-value between 0.01 and 0.05; NS, nonsignificant. (**C**) Summary of the assays, including the results of survival curve comparisons (*p*-values) between mix (coinfection) and the individual strain that exhibited the fastest fly killing, and results of the inoculum ratios of the suspensions used for infection (strain 1: strain 2, targeted ratio of 1:1). Color indicates the type of effect related to the polymicrobial condition on infection outcome with the fastest fly killer: red, synergistic effect; blue, antagonistic effect; grey, no effect (neutral). (**D**) Bacterial counts and median counts (CFU/fly) at H6 of flies infected with single strain (blue) and mix (red).

**Figure 2 pathogens-12-00405-f002:**
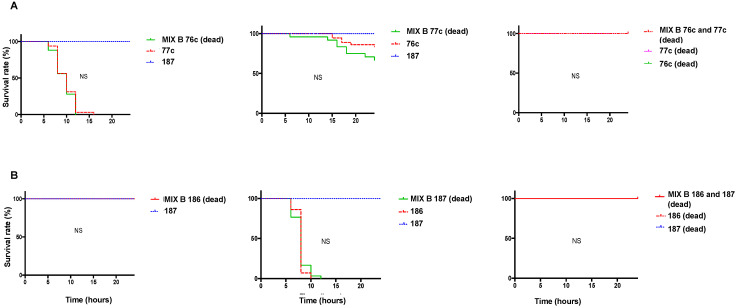
Survival curves of wild-type (WT, w1118) *Drosophila melanogaster* infected with individual strains and with mix comprised of one living strain and one dead strain (left and middle) or two dead bacteria (right). (**A**) Strains 76c ± 77c, (**B**) strains 186 ± 187. The survival curve of the coinfection was compared to that of the infection with the individual strain that exhibited the fastest fly killing. Survival curves representative of at least three independent experiments of the strains tested individually (dotted lines) and in pairs (solid lines), *n* = 30 flies per condition, ratio of 1:1. Bacteria from mix made of two dead bacteria were grown together. NS, nonsignificant.

**Figure 3 pathogens-12-00405-f003:**
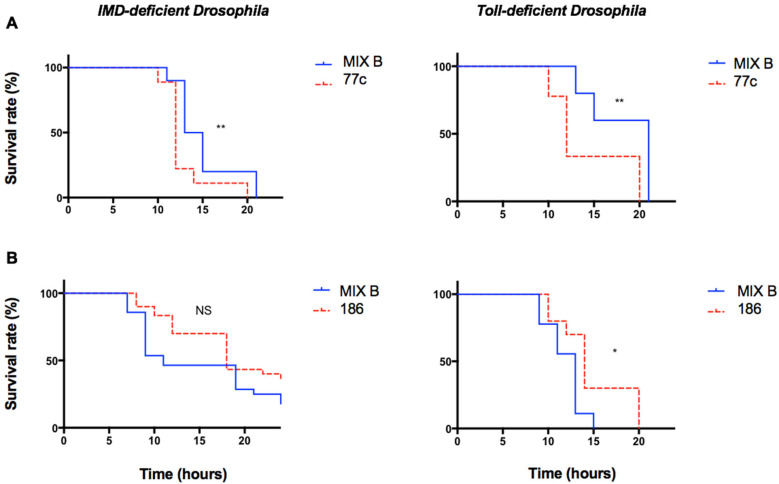
Survival curves of different mutant lineages of *Drosophila melanogaster* infected with individual strains and with mix. The survival curve of the coinfection was compared to that of the infection with the individual strain that exhibited the fastest fly killing. Infected flies belonged to two different lineages: Myd88 mutant flies (Toll deficient) and Relish mutant flies (IMD deficient). Survival curves representative of at least three independent experiments of the strains tested individually (dotted lines) and in pairs (solid lines), *n* = 30 flies per condition, ratio of 1:1. Experiments were carried out with a mix B inoculum. (**A**) Strains 76c ± 77c, (**B**) strains 186 ± 187. ** *p*-value between 0.01 and 0.009, * *p*-value between 0.01 and 0.05.

## Data Availability

The data that support the findings of this study are available from the corresponding author (AR and BL) upon reasonable request.

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
