# Peer review of "Drosophila melanogaster Systemic Infection Model to Study Altered Virulence during Polymicrobial Infection by Aeromonas"

_pathogens, 2023, doi:10.3390/pathogens12030405_

Round 1

Reviewer 1 Report

The study by Robert A et al, titled “Drosophila melanogaster systemic infection model to study altered virulence during polymicrobial infection by Aeromonas”, developed a drosophila model for polymicrobial infection. This study is too preliminary for publication.

Specific comments

There is no novelty in this study, and they have not used vehicle control in all the survival experiments.

Figure 1B. Author injected flies with three clinical pairs of strains- Mix A (25a + 25b) and Mix B (186+ 187) (76c + 77c). They have reported that 25a + 25b stains exhibited similar virulence phenotypes when tested in pairs or individually, 186+ 187 pair have synergistic action and 76c + 77c pairs are antagonistic. – I don’t understand why author injected different bacterial numbers ie 25a + 25b pair (500 CFU/fly), 186+ 187 pair (1000 CFU/fly), and 76c + 77c pair (1000 CFU/fly).

Also, author need to use low bacterial number (100 CFU/fly or 200 CFU/fly) for infections and check survival for 7 days.

Author showed total changes of bacterial number in response to pairs or individual stain injection. – Since this study is on polymicrobial infection model, this information is not sufficient. Author should show the effect of pairs or individual stain on bacterial diversity and author should show CFU changes in different parts (Gut, hemolymph, and fat body)

Author showed 186+ 187 pairs have synergistic effect. If this argument is correct, why total CFU/fly of is low in response to 186+ 187 pairs when compare 187 alone?

I am glad that logic of using IMD or toll mutants but need to confirm by using RNAi lines.

Author Response

We thank Reviewer #1 for these comments. We are sorry to read these comments are opposite to those of Reviewers #2 and #3 who assessed that “experimental plan was sound. Experiments were adequately controlled” (Reviewer #2); or that “The study setting and the topic is interesting, and we lack data on infection dynamics in polymicrobial infections” (Reviewer #3). 

Some specific comments are difficult to follow, e.g., when information is already available in the manuscript or discussed (e.g., inoculum number comments) and unremarkable to other reviewers.

We also thank Reviewer #1 for the suggested experiments. However, it seems to us that they are out of the scope of this study. This study is primarily aimed at showing that this model can distinguish between different effects produced by different paired strains recovered from acute infection diseases, which we show. For instance, the question of bacterial loads in various anatomic zones in the fly or the point on reinforcing evidence of the no-role of TOLL and IMD pathways using RNAi are fascinating questions, but they go beyond the aim of the current study that was to set up a model able to discriminate between different effects during polymicrobial infection. These questions correspond to a further study with different aims (e.g., how to explain the observed effect). Instead, we revised the discussion to include points that were not already discussed in the discussion (e.g., RNAi).

Reviewer 2 Report

The authors present a study describing the use of a Drosophila model for studying polymicrobial infections. The work was well thought out and the experimental plan was sound. Experiments were adequately controlled. 

Minor comments:

Line 29: The authors are overreaching with the model's association with human infections. Specific correlational study data between the fly model and human infections was not presented. In the absence of more in-depth comparisons, the fly and human outcome data could be purely coincidental. The sentence should be reworded or removed.

Line 101: "6" should be spelled out. Also, some additional information on how the isolates related to the clinical disease conditions would be helpful. For example, were isolates 76c/77c found to be the cause of the diarrhea described in Table 1?

Line 105: Please provide the cryopreservation media used.

Line 106: the term "fresh" is arbitrary, please provide the incubation time the colonies were grown for instead.

Line 109: Suggest changing the sentence to read " Growth curves were performed..."

Line 111: "pair" should be "pairs"

Line 112: It may be helpful to reference Table 1 at the end of the sentence.

Line 123: Centrifuge speeds should be listed in x g not rpm.

Lines 130-138: Seems a bit long, I would suggest breaking this into two sentences. 

Line 131: "infection" should be infections

Line 132: Suggest changing to "... same total amount of bacteria (all cells cumulated) compared to single-strain suspensions..."

Line 135: "infection" should be infections.

Line 136: A space is needed between E. and coli and both should be italicized. 

Line 138: State that these are control infections.

Line 143: Should be two sentences, "...conditions. Flys were infected..."

Lines 154-161: Some more information is needed about how the antagonistic/synergistic/neutral effect determinations were made. Specifically, were the averages of all replicates used to make the determinations. Perhaps adding this information into the statistical analysis section would be appropriate. 

Line 166: Define PBS before abbreviating.

Line 167: Consider changing "aerobic atmosphere" to "aerobic conditions"

Line 169: It is unclear what mixed conditions refers to.

Line 172-174: Was this method of identifying colonies experimentally confirmed? For example, were a subset of phenotypically identified colonies from the experimental samples independently verified by a second method to ensure that the morphologic method produced the correct colony identification. 

Line 214: "As expected" is not needed here, can just say "The E. coli strain 187..."

Line 230: Do the survival curves represent the average of the 30 flys?

Line 245: "10-4" should be superscript? 

Lines 246-249: Were the two conditions compared statistically? If so, it would be helpful to state the outcome of the statistical analysis.

Line 254: Numbers less than ten should be spelled out.

Line 255: "In contrast and interestingly..." should be either "In contrast" or "interestingly" but both are not needed.

Line 257: 10-4 should be superscript?

Line 271 and 275: E. coli should be italicized.

Lines 277-281: This information is stated in the methods section and doesn't need to be restated here. Consider removing.

Line 283: State if "representative" refers to a single experiment or the average of the three independent experiments. 

Line 331: Spell out numbers less than ten.

Lines 337-341: The figure shows the 50% survival rate for the 76b/77c mixture was achieved past ten hours in both IMD and Toll-deficient strains whereas the 50% survival rate in the wild type flys was prior to ten hours. Do the authors think this delay is relevant? Please discuss.

Line 366-367: Additional context should be provided concerning the statement about the effects being consistent with clinical severity observed in humans. It does not appear that the authors performed a rigorous comparison between the fly and human outcomes (if so, this should be stated in the results section). Therefore, the observation may be coincidental especially considering there were only three humans sampled. There are several factors that can affect the course of diarrheal infections in humans including microbial community, patient age, digestive tract health, etc. If the comparisons to human infections were not tested directly, the authors should state that while the observations may be consistent with the severity observed in humans they were not specifically tested in this study and may also be coincidental.

Line 376: Looks like "depending" should be removed. 

Line 386: "inevitable" is not needed, consider removing.

Line 392: See above comments about correlations with human disease. Perhaps change to "...may be consistent with clinical severity." 

Line 412 and 438: Italicize organism names

Author Response

Response: We agree with Reviewer 2. The sentence was removed from the abstract (line 29). In addition, this point was mitigated throughout the manuscript by either deleting sentence or  some rephrasing (e.g., lines 350-351).

Line 101: "6" should be spelled out. Also, some additional information on how the isolates related to the clinical disease conditions would be helpful. For example, were isolates 76c/77c found to be the cause of the diarrhea described in Table 1?

Response: Information to relate these strains to the clinical condition considered several information, e.g. site of infection, clinical status and evolution, and the physician decision to treat. One case (25a/25b) was reported in a larger series that was deeply investigated on the clinical aspects and published (Lamy et al, 2008, ref 38), and clinician decided not to treat, with further good evolution. Opposite, case infection 186/187 was a bloodstream infection with clinician decision to treat using a treatment to cover the two bacteria.

Concerning the infection caused by 76c/77c: as indicated in Table 1, isolates 76c and 77c were recovered from a 5-year-old child who showed 24h before being admitted to the hospital, fever (38°C) and abdominal pain. At the exploration, fever was 38.9 C presented apparently normal depositions. Considering the symptomatology and, the underlaying condition of the patient (he have been operated of myelomeningocele with hydrocephalus due to Chiari malformation) the pediatrician ordered the coproculture to determine if symptoms could be attributed to an infectious diarrhea.  The stool coproculture was screened for Salmonella, Shigella, Campylobacter, Yersinia, Aeromonas, Vibrio and diarrheic E. coli using the standard microbiological techniques and biochemical testing. According to our records, growth was only observed in the cefsulodin irgasan novobiocin (CIN) agar medium and of the two colonies selected for  identification  (76c and 77c) were identified with Vitek phenotypic identification system. The system identified 2 Aeromonas that belonged to 2 different species. Meanwhile (i.e., before obtaining the stool culture results), the patient was administered a broad spectrum antimicrobial treatment, vancomycin and gentamicin. It is therefore unclear whether the diarrhea resolved spontaneously or because of the antimicrobial treatment, but the pediatrician considered the strains were responsible of the symptoms. The patient suffered a co-occurring infection: after feces were taken on the same day of arrival, 19/8/92, on the 20/8/92 clinicians discovered the tip of the catheter had pus (after culture S. pyogenes), the device was changed, the chirurgical procedure was revised and a new chirurgic procedure 4/9/92 was done installing a valve. The post operatory evolution was good and was relished from the hospital on 12/9/92.

In fact, MJ Figueras et al have described in previous publications that 2-20% of the diarrheal cases from children around the word have been induced solely by species of Aeromonas; recent studies, like the study performed by Shrestha et al. (2022) that investigated 1,200 children with acute diarrhoea (cases) and 1,200 children without diarrhoea (control subjects) from 3 to 60 months of age and demonstrated that Aeromonas were identified significantly more frequently (9% versus 3%) in cases than control subjects.

Posterior molecular genotyping and identification methods (ERIC-PCR, sequences of the rpoD gene and whole genome sequences) confirmed both isolates as different strains belonging to two different species A. media (76c) and A. veronii (77c) as they appear in the study of Mosser et al. (2015). In a posterior study  by Talagrand-Reboul et  al (2017) who analysed the sequence of the complete genomes, it was recognised that strain 76c to belong to the novel  species A. rivipollensis and strain 77c was confirmed to be A. veronii.

We revised Table 1 by adding i) some references, iii) more information (e.g., fever, decision to decision to treat, ii) adding a new column entitled ‘comments’ with more information, iii) a footnote explaining the fact that the strain 76c was identified under different names.

References:

Shrestha SK, Shrestha J, Mason CJ, Sornsakrin S, Dhakhwa JR, Shrestha BR, Sakha B, Rana JC, Srijan A, Serichantalergs O, Sethabutr O, Demons S, Bodhidatta L. Etiology of Acute Diarrheal Disease and Antimicrobial Susceptibility Pattern in Children Younger Than 5 Years Old in Nepal. Am J Trop Med Hyg. 2022 Dec 5;108(1):174-180.

Mosser T, Talagrand-Reboul E, Colston SM, Graf J, Figueras MJ, Jumas-Bilak E, et al. Exposure to pairs of Aeromonas strains enhances virulence in the Caenorhabditis elegans infection model. Front Microbiol. 2015;6:1218.
Talagrand-Reboul E, Roger F, Kimper JL, Colston SM, Graf J, Latif-Eugenín F, Figueras MJ, Petit F, Marchandin H, Jumas-Bilak E, Lamy B. Delineation of Taxonomic Species within Complex of Species: 
Aeromonas media and Related Species as a Test Case. Front Microbiol. 2017 Apr 18;8:621.

Line 105: Please provide the cryopreservation media used.

Response: done (line 107-108)

Line 106: the term "fresh" is arbitrary, please provide the incubation time the colonies were grown for instead.

Response: this point is indeed specified in the previous sentence (overnight). We revised the sentence to clarify and delete ‘fresh” (please see line 109)

Line 109: Suggest changing the sentence to read " Growth curves were performed..."

Response: revised (line 112)

Line 111: "pair" should be "pairs"

Response: revised  (line 114)

Line 112: It may be helpful to reference Table 1 at the end of the sentence (please see line113 in the revised manuscript).

Response: done (line 115)

Line 123: Centrifuge speeds should be listed in x g not rpm.

Response: revised  (line 127)

Lines 130-138: Seems a bit long, I would suggest breaking this into two sentences. 

Response: the sentence was revised accordingly (lines 133-141).

Line 131: "infection" should be infections

Response: revised (line 134)

Line 132: Suggest changing to "... same total amount of bacteria (all cells cumulated) compared to single-strain suspensions..."

Response: revised as suggested (line 134-136)

Line 135: "infection" should be infections.

Response: revised as suggested (line 138)

Line 136: A space is needed between E. and coli and both should be italicized. 

Response: revised as suggested (line 139)

Line 138: State that these are control infections.

Response: revised as suggested (line 143)

Line 143: Should be two sentences, "...conditions. Flys were infected..."

Response: revised as suggested  (line 146)

Lines 154-161: Some more information is needed about how the antagonistic/synergistic/neutral effect determinations were made. Specifically, were the averages of all replicates used to make the determinations. Perhaps adding this information into the statistical analysis section would be appropriate. 

Response: We did not average results of all replicates. To determine effect, we proceeded as follows: an effect was considered as antagonistic/synergic when i) survival curve of the mix was statistically different from that of the monomicrobial infection, according to definitions given in lines 158-166 (i.e., flies dying more slowly/faster, respectively), and ii) the effect was reproducible, i.e. observed from at least 3 independent assays. Results of these 2 points are shown in Figure 1B and 1C, respectively. A neutral effect was determined when survival curve comparison was statistically and reproducibly not different (3 independent assays). Comparisons were performed between the curve of the mix and that of the individual strain that exhibited the fastest fly killing, as already specified in the statistical analysis section (lines 200-202).

To clarify, we revised the manuscript by adding the following sentence in the statistical analysis section:

“Curve comparisons were independently analysed for every assay. Antagonism or synergy was determined when i) survival curve with the mixed infection was found significantly slower or faster compared to that of the monomicrobial infection (hereafter effect), respectively, and when ii) this effect was reproducible, i.e., when it was observed from at least 3 independent assays.” (lines 202-207)

Line 166: Define PBS before abbreviating

Response: revised as suggested  (line 170)

Line 167: Consider changing "aerobic atmosphere" to "aerobic conditions"

Response: revised as suggested  (line 172)

Line 169: It is unclear what mixed conditions refers to.

Response: we revised the sentence to clarify this point  (line 173-174)

Line 172-174: Was this method of identifying colonies experimentally confirmed? For example, were a subset of phenotypically identified colonies from the experimental samples independently verified by a second method to ensure that the morphologic method produced the correct colony identification. 

Response: the method was verified using 2 strategies. First, we verified the method by identifying several colonies using the matrix-assisted laser desorption ionization–time of flight mass spectrometry (MALDI-TOF MS) using a Microflex LT (Bruker, Wissembourg, France) with MALDI Biotyper 3.1 software and the Bruker database 7311 (Bruker). Second, we systematically cultured the single-strain suspensions that were used as controls for phenotypic aspects.  We revised the legend of supplementary Figure S2, as follows:

“Accuracy of the colony recognition was previously verified using the matrix-assisted laser desorption ionization–time of flight mass spectrometry (MALDI-TOF MS) using a Microflex LT (Bruker, Wissembourg, France) with MALDI Biotyper 3.1 software and the Bruker database 7311 (Bruker).” (line 191-193)

Line 214: "As expected" is not needed here, can just say "The E. coli strain 187..."

Response: revised as suggested (line 226).

Line 230: Do the survival curves represent the average of the 30 flys?

Response: each survival curve summarizes the outcome of the 30 flies infected in each condition as a function of number of dead flies at the time the flies died. As survival is presented in percent survival, numbers of flies do not appear directly. Neither average dead flies is presented nor average curves. Survival curves presented in supplementary Figure S3 are representative of at least three independent experiments. To clarify a possible confusion, we revised the figure legend. –(line 239-240)

Line 245: "10-4" should be superscript? 

Response: yes, sorry for this miss. We revised (lines 252 and 255)

Lines 246-249: Were the two conditions compared statistically? If so, it would be helpful to state the outcome of the statistical analysis.

Response: survival curves were compared within each assay (i.e., comparisons between the curve of the mix and that of the individual strain that exhibited the fastest fly killing) but not between assays due to some additional sources of variability (e.g., assays performed on different days, thus with different batch of flies). For a given pair, analyses showed consistent difference between curves with the 2 assays, e.g., slower killing for pair 76c/77c. The sentence in the manuscript “We observed similar effects when flies were infected with either mix A or mix B for pairs” reflects these observations. However, this adds little to the manuscript and is rather confusing. We clarified by deleting this sentence.

Line 254: Numbers less than ten should be spelled out.

Response: revised accordingly (line 263).

Line 255: "In contrast a

nd interestingly..." should be either "In contrast" or "interestingly" but both are not needed.

Response: We revised the manuscript. We changed for “second” to also address another comment (from another reviewer)  (line 250).

Line 257: 10-4 should be superscript?

Response: revised (line 252).

Line 271 and 275: E. coli should be italicized.

Response: revised (line 272 and 276).

Lines 277-281: This information is stated in the methods section and doesn't need to be restated here. Consider removing.

Response: we removed this piece of information and instead referred to the method section (line 277)

Line 283: State if "representative" refers to a single experiment or the average of the three independent experiments. 

Response: this piece of information was specified (line 279-280)

Line 331: Spell out numbers less than ten.

Response: revised throughout the manuscript

Lines 337-341: The figure shows the 50% survival rate for the 76b/77c mixture was achieved past ten hours in both IMD and Toll-deficient strains whereas the 50% survival rate in the wild type flys was prior to ten hours. Do the authors think this delay is relevant? Please discuss.

Response: the fly lineages we used for wild type and the mutant flies have the same genetic background (w1118), but the w1118 and the mutant flies do not originate from the same lab, and these lineages have not been backcrossed. Consequently, results from Figure 1 (WT) and Figure 3 (Toll and IMD mutants) cannot and should not be compared. That is why we presented results on 2 separate figures and why Figure 3 did not include any data with WT flies.

Line 366-367: Additional context should be provided concerning the statement about the effects being consistent with clinical severity observed in humans. It does not appear that the authors performed a rigorous comparison between the fly and human outcomes (if so, this should be stated in the results section). Therefore, the observation may be coincidental especially considering there were only three humans sampled. There are several factors that can affect the course of diarrheal infections in humans including microbial community, patient age, digestive tract health, etc. If the comparisons to human infections were not tested directly, the authors should state that while the observations may be consistent with the severity observed in humans they were not specifically tested in this study and may also be coincidental.

Response: we agree with Reviewer 2. We mitigated this point and we revised the manuscript by adding a sentence (line 350-351)

Line 376: Looks like "depending" should be removed. 

Response: revised as suggested (line 364)

Line 386: "inevitable" is not needed, consider removing.

Response: revised as suggested (line 381)

Line 392: Seeabove comments about correlations with human disease. Perhaps change to "...may be consistent with clinical severity." 

Response: this sentence was deleted

Line 412 and 438: Italicize organism names

Response: apology for this miss. Revised as suggested (lines 399 and 426)

Reviewer 3 Report

The authors have infected fruit flies with a mix of two bacterial strains, mainly aeromonads, and observed the pathogenicity of the mix in flies as compared to that of single infections by these strains. They could observe various kinds of effects (synergetic, antagonist, neutral), which were modulated when two strains (instead on one strain) were inoculated to flies simultaneously. The study setting and the topic is interesting, and we lack data on infection dynamics in polymicrobial infections. There are several interesting observations in this study. However, the generalization of the model based on the study with 3 bacterial pairs does not seem reliable. It remains vague what the authors exactly mean by the model. In addition, the discussion is at very general level and sometimes cannot be followed. More traditional structure of the manuscript with no discussion sentences in the results, as well as discussion section in which the results are interpreted/discussed and compared with other similar studies, would be recommended. Below, specific comments are given for further improvement of the manuscript.

L19 ‘worse’ than which?

L21 ‘We’ instead of ‘we’

L44-45 The sentence starting ‘Polymicrobial infections encompass…’ is difficult to follow. Please, clarify.

L67 What does mean that ‘model is long’?

L79-83 The sentence is difficult to understand as such. Please, modify and clarify.

L102 ‘Another’ is not a suitable word here.

L104 Is it possible to give a genbank accession number?

L120 What does mean ‘infections were adapted from Lee..’ Please, open more.

L121 is it 108 or 10^8?

L124 Use g-value instead of rpm 

L150 Can the authors add more data on Imler (for instance Institute)?

L150 Past tense (corresponded) throughout the paragraph. Also, L176 started dying

L217 The sentence is a bit clumsy, please, clarify.

S3. ‘…infected by …infection. Please, correct.

L245-246 The statement should be proven more strongly and with more strains. More evidence is required. Maybe some scoring system or similar for the clinical symptoms could be set up?

L296 Do the author mean really one dead bacterium?

L346 Which tenet do the authors mean?

L346-8 The sentence remains unclear, please, clarify.

L387 Please, open a bit more. It is not generally known which variability is meant here.

L388-92 Please, clarify these sentences. The text cannot be easily followed.

L400-1 In which way the model was useful in a flap infection?

L403 The sentence remains unclear, please, modify. L 403-6 the text is quite difficult to follow. Please, clarify. It could also be reconsidered if the text is suitable for a discussion section. Also, L418-24.

L411 and 423 Why are the reference numbers marked in bold?

L416 The authors could discuss a bit more about Sibley study. Was their study done using the same bacterial strains, was the same lineage of flies used?

L428 Please, open more what ‘well-defined’ means in this context.

L436 The sentence starting with ‘One..’. Do the authors mean ‘one study’? Or perhaps something else?

Author Response

The authors have infected fruit flies with a mix of two bacterial strains, mainly aeromonads, and observed the pathogenicity of the mix in flies as compared to that of single infections by these strains. They could observe various kinds of effects (synergetic, antagonist, neutral), which were modulated when two strains (instead on one strain) were inoculated to flies simultaneously. The study setting and the topic is interesting, and we lack data on infection dynamics in polymicrobial infections. There are several interesting observations in this study. However, the generalization of the model based on the study with 3 bacterial pairs does not seem reliable. It remains vague what the authors exactly mean by the model. In addition, the discussion is at very general level and sometimes cannot be followed. More traditional structure of the manuscript with no discussion sentences in the results, as well as discussion section in which the results are interpreted/discussed and compared with other similar studies, would be recommended. Below, specific comments are given for further improvement of the manuscript.

Response: We thank reviewer #3 for these comments. To address them we revised the manuscript by revising the result and the discussion sections:

  • We deleted from the result section several sentences that were moved in the discussion section, i.e. (moved into discussion, lines 352-354 and 356-361).

We revised the text throughout discussion and deleted several general points. However, we chose to keep several points that could be perceived as general by the reviewer but that would be important to a reader who would not be familiar to either the Drosophila infection model or to interactions that occur during polymicrobial infection. In addition, we did not want to change too deeply a discussion that was not issuing to other reviewer. Finally, we complemented the discussion by comparing our results with some other studies (e.g., lines 403-408)

L19 ‘worse’ than which?

Response: the abstract was revised to specify comparison with monomicrobial infections (line 20)

L21 ‘We’ instead of ‘we’

Response: revised (line 21)

L44-45 The sentence starting ‘Polymicrobial infections encompass…’ is difficult to follow. Please, clarify.

Response: this sentence was revised (line 44-46)

L67 What does mean that ‘model is long’?

Response: We meant the models have long time to results. The sentence was revised (line 68)

L79-83 The sentence is difficult to understand as such. Please, modify and clarify.

Response: the sentence was revised (line 80-84)

L102 ‘Another’ is not a suitable word here.

Response: revised (line 103)

L104 Is it possible to give a genbank accession number?

Response: genome sequences were deposited in the European Nucleotide Archive
(ENA) database (not in genbank). Accession numbers were added in the revised manuscript (lines 105-107).

L120 What does mean ‘infections were adapted from Lee..’ Please, open more.

Response: we agree that this sentence is clumsy and confusing. To fully address this concern, we removed this sentence as it is inappropriately referenced here.

L121 is it 108 or 10^8?

Response: revised (line 124)

L124 Use g-value instead of rpm 

Response: revised (line 127)

L150 Can the authors add more data on Imler (for instance Institute)?

Response: the manuscript was revised accordingly (lines 153-154)

L150 Past tense (corresponded) throughout the paragraph. Also, L176 started dying

Response: revised  (lines 154)

L217 The sentence is a bit clumsy, please, clarify.

 Response: during the revision of the manuscript, with sentence was revised and moved to the discussion section  (lines 354)

S3. ‘…infected by …infection. Please, correct.

Response: revised  (lines 238)

L245-246 The statement should be proven more strongly and with more strains. More evidence is required. Maybe some scoring system or similar for the clinical symptoms could be set up?

Response: the sentence was deleted from the revised manuscript.

L296 Do the author mean really one dead bacterium?

Response: we are unsure to properly understand this comment. However, we understand that using the word bacterium may be confusing. We revised the sentence by replacing bacterium  by strain  (lines 293)

L346 Which tenet do the authors mean?

Response: the sentence was unclear and we revised it  (lines 335)

L346-8 The sentence remains unclear, please, clarify.

Response: the sentence was revised (lines 335-337)

L387 Please, open a bit more. It is not generally known which variability is meant here.

Response: we revised the sentence to better specify which variability we are discussing about (lines 382-383)

L388-92 Please, clarify these sentences. The text cannot be easily followed.

Response: we rephrased the text to clarify (lines 383-386)

L400-1 In which way the model was useful in a flap infection?

Response: We detailed how this model helped to investigate this infection. We also revised ‘useful’ as we reckon that the word ‘used’ would be indeed more appropriate.  (lines 369 and after)

L403 The sentence remains unclear, please, modify. L 403-6 the text is quite difficult to follow. Please, clarify. It could also be reconsidered if the text is suitable for a discussion section. Also, L418-24.

Response: we revised this paragraph to clarify (lines 393-395) and we deleted several parts

L411 and 423 Why are the reference numbers marked in bold?

Response: there is no reason for marking references in bold. This inconsistency was corrected    (lines 402 and deleted)

L416 The authors could discuss a bit more about Sibley study. Was their study done using the same bacterial strains, was the same lineage of flies used?

Response: we expanded about Sibley study (lines 403-411)

L428 Please, open more what ‘well-defined’ means in this context.

Response: the well-defined conditions are indeed detailed in initial manuscript in the two sentences that follow. We understand by reading this comment that this is unclear to the reader. We revised the manuscript by rephrasing the companion sentences lines 415-419.

L436 The sentence starting with ‘One..’. Do the authors mean ‘one study’? Or perhaps something else?

Response: we revised the sentence by adding “study”    (lines 424)

Round 2

Reviewer 1 Report

Authors have not addressed any comments and the available data is not sufficient for publication.  

Author Response

Noted

Reviewer 3 Report

The authors have answered to the comments and improved the manuscript. Some, mainly grammatical or technical corrections have appeared in the new sentences. Below, some comments are given for finalizing the manuscript.

L20 check ‘microbia linfections’

L135 Check if the explanation in the brackets in the end of the sentence is necessary anymore.

Table 1 there are some sentences (first words) in the Comments column that start with a small letter, please, correct

Footnote: ‘latter’ or ‘’later’, or something else? This is not quite clear.

L266 Check ‘0’

L337-340 The sentence ‘Model organisms have emerged as … in vivo models…, of which the fruit fly model’ remains still unclear. Please, try to explain using some other words.

L340 Check this line. Maybe the title is in incorrect place.

L356 Perhaps ‘this mutant data obtained’ (instead of mutant data) if the authors mean their mutant flies used in this study.

L358 ‘…RNAi lines of these flies…’? Would that be correct?

L370-7 The sentence could start with ‘Previously, the fly model was used…’, for clarity (if I understood correctly). In addition, did the infection happen after leech therapy of the heart flap? If so, the leach therapy could be mentioned in the first sentence, otherwise it is difficult to understand the following sentence. Also, the ‘two aeromonad isolates’ instead of two isolates (if I understood correctly). 

L393 perhaps ‘another microbe’ instead of ‘other microbe’ (or ‘other microbes’)

Author Response

We thank reviewer #3 for these comments. We revised the manuscript as follows to address these comments. We hope that the manuscript will be suitable for publication.

L20 check ‘microbia linfections’

Response: We revised the sentence and deleted the typo (L20)

L135 Check if the explanation in the brackets in the end of the sentence is necessary anymore.

Response: We deleted it (L135)

Table 1 there are some sentences (first words) in the Comments column that start with a small letter, please, correct

Response: these points were revised (L233-234)

Footnote: ‘latter’ or ‘’later’, or something else? This is not quite clear.

Response: later is more appropriate, and we revised the manuscript (L234)

L266 Check ‘0’

Response: deleted (L266)

L337-340 The sentence ‘Model organisms have emerged as … in vivo models…, of which the fruit fly model’ remains still unclear. Please, try to explain using some other words.

Response: we changed for : “Model organisms have emerged as relevant in vivo models to study infection pathophysiology of polymicrobial infections (11,12,33,34), of which the fruit fly model. Our results add to the existing evidence the fly model is a promising model to study polymicrobial infection” (L335-338)

L340 Check this line. Maybe the title is in incorrect place.

Response: thanks for this point. The manuscript was revised (L339)

L356 Perhaps ‘this mutant data obtained’ (instead of mutant data) if the authors mean their mutant flies used in this study.

Response: we revised the sentence (L355)

L358 ‘…RNAi lines of these flies…’? Would that be correct?

Response: Yes, we changed for it. (L356)

L370-7 The sentence could start with ‘Previously, the fly model was used…’, for clarity (if I understood correctly). In addition, did the infection happen after leech therapy of the heart flap? If so, the leach therapy could be mentioned in the first sentence, otherwise it is difficult to understand the following sentence. Also, the ‘two aeromonad isolates’ instead of two isolates (if I understood correctly). 

Response: We revised to clarify this sentence. (L369-371)

L393 perhaps ‘another microbe’ instead of ‘other microbe’ (or ‘other microbes’)

Response: revised. (L393)
